# Internet of things (IoT) for smart agriculture: Assembling and assessment of a low-cost IoT system for polytunnels

Nuwan Jaliyagoda[1], Sandali Lokuge[2], P. M. P. C. Gunathilake[3], K. S. P. Amaratunga[4], W. A. P. Weerakkody[5], Pradeepa C. G. Bandaranayake[2]*, Asitha U. Bandaranayake[1]*

1 Department of Computer Engineering, Faculty of Engineering, University of Peradeniya, Peradeniya, Sri Lanka, 2 Agricultural Biotechnology Centre, Faculty of Agriculture, University of Peradeniya, Peradeniya, Sri Lanka, 3 Department of Statistics & Computer Science, Faculty of Science, University of Peradeniya, Peradeniya, Sri Lanka, 4 Department of Agricultural Engineering, Faculty of Agriculture, University of Peradeniya, Peradeniya, Sri Lanka, 5 Department of Crop Science, Faculty of Agriculture, University of Peradeniya, Peradeniya, Sri Lanka

* pradeepag@agri.pdn.ac.lk (PCGB); asithab@eng.pdn.ac.lk (AUB)

**Data Availability Statement:** The data used for this research will be available from the Institutional Data Repository, http://agbc-fe.pdn.ac.lk/datasets/

## Abstract

Internet of things (IoT) applications in smart agricultural systems vary from monitoring climate conditions, automating irrigation systems, greenhouse automation, crop monitoring and management, and crop prediction, up to end-to-end autonomous farm management systems. One of the main challenges to the advancement of IoT systems for the agricultural domain is the lack of training data under operational environmental conditions. Most of the current designs are based on simulations and artificially generated data. Therefore, the essential first step is studying and understanding the finely tuned and highly sensitive mechanism plants have developed to sense, respond, and adapt to changes in their environment, and their behavior under field and controlled systems. Therefore, this study was designed to achieve two specific objectives; to develop low-cost IoT components from basic building blocks, and to study the performance of the developed systems, and generate real-time experimental data, with and without plants. Low-cost IoT devices developed locally were used to convert existing basic polytunnels to semi-controlled and monitoring-only polytunnels. Their performances were analyzed and compared with each other based on several matrices while maintaining the planted tomato variety and agronomic practices similar. The developed system performed as expected suggesting the possibility of commercial applications and research purposes.

## Introduction

The Internet of Things, IoT [1] is a network of devices embedded with electronics, sensors, actuators, software, and connectivity, allowing these devices to connect, gather, and exchange data, take actions based on these data, and learn from those data towards future improvements. Like in many other domains, IoT technology has brought about the next revolution in Agriculture [2, 3]. IoT in smart or precision agricultural systems already plays a critical role in

greenhouses/v1/ and from code repository, https://github.com/cepdnaclk/Smart-Agriculture-a-low-cost-IoT-system-for-polytunnels.

**Funding:** The Early Career Fellowship of the Organization for Women in Science for Developing World (OWSD, Early Career Fellowship https://owsd.net/) funded this project – the award agreement 4500406736 was awarded to Pradeepa C.G. Bandaranayake. The principal investigator acknowledges the immense support and flexibility of the OWSD Early Career Fellowship program and the OWSD staff based at the offices of The World Academy of Sciences (TWAS), in Trieste, Italy.

**Competing interests:** The authors have declared that no competing interests exist.

monitoring crop growth, selection of fertilizer, early detection of diseases, irrigation decision support systems, etc., especially in developed countries and high-end markets [4–6]. The continuous increase in the global population results in an increased need for food while reducing the arable land and water required for farming. Therefore, to realize the sustainable development goals of agriculture, it is paramount that the benefits of smart agriculture become commonplace, mainly in view of the efficient use of resources while increasing production [7].

Generally, the agricultural IoT can be divided into five main layers: the physical or perception layer, the network layer, the middleware layer, the service layer, and the application layer [8, 9]. Together these layers are mainly responsible for data acquisition, data transmission, data storage data analysis, data presentation, and related applications. Of them, the physical layer is the bottom layer of the system, which is mainly responsible for data collection. It consists of many sensor nodes, actuators, and one or more sink nodes, which act as data collection points. Each sensor node may consist of several sensors to sense variables such as humidity, light, and temperature with a microcontroller and wired/wireless communication module to facilitate data transmission [10]. These end devices could be powered by batteries or renewable energy sources, which significantly impact the sensor network's lifetime. The sensor nodes and actuators transmit data or receive commands, respectively, mainly through wireless communication technology, which operates in the network layer. On the one hand, the network layer receives the data collected by the sensor layer and uses the local network, and maybe even the Internet, to transmit the data to upper layers for storage and further processing [11]. On the other hand, the control commands for the sensors and actuators generated by the user applications are also passed through the network layer.

The wireless technologies that are used can range from common Wi-Fi, 4G/5G, and Bluetooth, to very specific technologies such as 6LowPAN, LoRa, and NB-IoT (Narrowband IoT). IoT's middleware and service layers emerged much later than the other three layers, which are necessary components of any IoT solution. The middleware layer is generally a software system that acts as an interface between IoT devices and applications [12]. The middleware layer aims to hide the hardware diversity and complexities from IoT applications and services, significantly increasing the development and usability of IoT applications in every domain. The middleware layer mainly aggregates and processes the data generated by the IoT devices hiding their heterogeneity, while providing a variety of context-based services to above to facilitate any type of application [13]. The service layer, on the other hand, is designed to provide services that are common to all IoT applications, including anomaly detection, data storage, diagnostics, intelligent decision-making, predictions, early warnings, etc. The service layer may utilize many technologies such as cloud computing, fog & edge computing, SDN/NFV, artificial intelligence, machine learning, and big data analytics to build a common platform that the applications can utilize [9]. The application layer is the upper layer of the system, which utilizes the services provided by the lower layers and a variety of IoT-specific messaging protocols, such as MQTT, XMPP, and CoAP, in order to perform a variety of domain-specific activities. Typically, an app or web-based remote monitoring platform will allow the users to log in to the system, obtain various types of environmental data, carry out configuration changes, or even directly control end devices such as fans, humidifiers, sprinklers, etc. Such platforms can easily provide many different visual data representations, including data trends over time and across different locations, even within a greenhouse. The users should be able to obtain two types of data: raw data, and preprocessed data. Users can choose the data source according to their own needs for subsequent data analysis and mining work. With all these commonplace technologies it is becoming possible to develop a deep and wide network of farming applications, to take sustainable farming to the next level, not only by automatically learning and benefiting from each other, but also by maximizing productivity, quality, and profitability [9]

Large-scale multinational companies have already produced such machinery and devices, and have built controlled environmental facilities on a variety of scales, mainly in developed countries. Most of these systems are quite expensive systems given that they are developed with state-of-the-art and top-of-the-line equipment, and the latest technology. Most of the time each of these parts is custom-built by different companies, while the integration is done by the solution provider. While such solutions could be affordable for high-end customers even in developing countries, these are still prohibitively expensive for researchers and farmers in those countries. In essence, most of the farmers in the world are nowhere near reaping the benefits of the latest innovations in this domain. More worryingly, these farmers are increasingly displaced by large-scale farming practices using the latest technology [14, 15].

In addition to not being able to afford the existing IoT solutions, the adaptation of similar, yet low-cost, IoT solutions in the agricultural domain is progressing slowly due to a lack of experimental data under operational environmental conditions. Most of the current designs are based on simulations and artificially generated data [16–18]. While there are applications of IoT technologies in field conditions there is a significant lack of experiments and resulting data that combines the effects of such environmental data with relevant agronomic data. Data and analysis from such isolated experiments make it unusable for farmers and end-users. Therefore, the current need is not only to develop low-cost low-power IoT solutions that are affordable to every farmer but also to make a comprehensive data set available so that they can make informed decisions.

In essence, agricultural production in controlled environments is increasingly feasible and new technologies in lighting, ventilation, robotics and irrigation are just a few of the innovations that enable the production of high-value specialty crops outside of a traditional field setting. Though such fully controlled systems are not affordable for developing countries, semi-controlled systems are feasible options. While several studies are done on changing one or two factors at a time, for example, Chlorophyll Fluorescence [19, 20], light and temperature range [21], light [22], and several others, no studies are reported on comprehensive analysis in dynamic IoT settings.

Considering the reasons detailed above, specifically on the non-affordability of smart agriculture solutions to researchers and farmers, and the lack of experimental data on field conditions to make informed decisions, this study was designed to achieve two specific objectives; 1) to develop low-cost IoT components from basic building blocks, and 2) to study the performance of the developed systems and generate real-time experimental data, with and without plants.

## Materials and methods

According to the current rules and regulations in the country, no specific permission or approvals were required from any organization to conduct this research.

### System architecture

**Monitoring.**   Monitoring plays a significant role in greenhouse climate control and such information is needed to take control decisions. We need to consider two things in monitoring environmental status. First, we need to decide from where the readings will be taken. For example, to measure atmosphere temperature, we can place the sensors anywhere in the 3D space of the greenhouse. Further, we need to consider how many sensors are to be set up, and where they should be located. The design factors are also important for cost optimization, to measure the environmental changes of the entire greenhouse using a minimum number of sensors.

Monitoring frequency is another important decision to be considered when taking sensor measurements. We don't want to miss the sudden climate changes happening inside the greenhouse by using a low frequency. If we go with a higher frequency, need to handle a lot of data, need large storage facilities to keep the data, and a lot of time needed to be spent on data handling and pre-processing.

**Controlling.** The main objective of control is to provide optimum environmental conditions for plant growth. The controlling actions can be taken from the received sensor readings and the time. In this research, we controlled the environmental temperature inside one greenhouse and compared the changes in environmental parameters with an uncontrolled greenhouse.

**Communication.** Controlling and Monitoring are the primary requirements of the system, but the communication between monitoring devices and controlling devices plays a major role. Either wired or wireless communication could be used for the purpose. By considering the cost, performance, and manageability, it was decided to use Wi-Fi as the primary communication medium between sensor/controller nodes and the cloud application during this research.

MQTT [23], a lightweight and publisher-subscriber-based protocol is used to communicate between nodes and web services, because of its low network bandwidth use and small code footprint.

## System design

The greenhouse monitoring and controlling system were designed in a modular way, in order to be easily customizable with different shaped and sized greenhouses. The system will contain 3 major components, 1) Sensor Nodes which collect environmental readings through various sensors 2) a Controller Unit that controls the environmental conditions, 3) a Web Server that stores collected data and let users see and manage the collected data and configure control conditions.

**Sensor node design.** The sensor node was designed with a microcontroller and electronic circuit, having the capability to connect several types of sensors that can collect the readings of the environmental conditions. In agriculture, crop yield mainly depends on temperature, relative humidity, and light intensity [24]. Therefore, those 3 parameters were selected for the initial stage of the monitoring. However, with the hardware availability, this sensor node can accommodate a maximum of any 8 digital sensors and any 4 analog sensors.

- 2 x DHT11 Temperature and Humidity sensor

- 1 x DHT22 Temperature and Humidity sensor

- 1 x BH1750 Light Intensity Sensor

DHT11 and DHT22 are commonly available and widely used temperature and humidity sensors [25, 26] with low cost, but relatively good accuracy and measuring range. BH1750 is a light intensity sensor that can measure the light intensity of the environment in lux. The physical properties of each sensor are available in Table 1.

The light sensor was in the main sensor unit, while the temperature/humidity sensors were separated from the main unit and placed at 3 different heights. The heights of those external sensors can be changed, to monitor the temperature and the humidity of the different heights. In the current study, we placed the external sensors at 10%, 35%, and 55% heights, and the light sensor at 65% from the ground to an average height of the greenhouse, as shown in Fig 1 (The heights were 0.4m, 1.2m, 2.0m, and 2.4m in numbers for this study)

**Table 1. The specifications of the sensors used in the control system.**

| Parameter | DHT11 | DHT22 | BH1750 |
|---|---|---|---|
| Sensor(s) | Temperature and Humidity | Temperature and Humidity | Light Intensity |
| Type | Digital | Digital | Digital |
| Communication | One-wire | One-wire | I2C |
| Measuring range | Humidity 20–90% RH; | Humidity 0–100% RH; | 0–27306 lx |
| | Temperature 0–50˚C | Temperature -40-80˚C | |
| Accuracy | Humidity ±4%RH | Humidity ±2%RH | 1.2 times |
| | (Max ±5%RH); | (Max ±5%RH); | |
| | Temperature ±2.0˚C | Temperature ±0.1˚C | |
| Resolution | Humidity 1%RH; Temperature 0.1˚C | Humidity 0.1%RH; Temperature 0.1˚C | 1 lx |

Heights of the temperature and humidity sensors were selected to fairly distribute the reading points within the greenhouse, with the estimated canopy height of the plants used for the experiments. The sensor nodes were located in each greenhouse as shown in Fig 2, in such a way that all sensors together give fair and distributed 3-dimensional coverage of temperature and humidity across the greenhouse, and 2-dimensional coverage of the light intensity received to the greenhouse. However, this arrangement subject to change based on the size, shape, and design of the greenhouse/polytunnel.

Apart from the main functional requirements, an RGB status indicator was included at the bottom of the sensor's main control unit to indicate the state of the sensor node as well as the error conditions. Different colors and blink counts were used to indicate different statuses. For example, two blinks of red color indicate an error with Sensor20, and one blink of blue color indicates the sensor readings were successfully sent to the web server

**Controller node design.** The controlling system was designed to control the environmental conditions inside the greenhouse. The controller designed for this study can control a blower motor, misting system, irrigation system, and side curtains of the greenhouse. However, it is possible to extend up to 8 controller devices with the current design.

The blower motor is a 3-phase 1.5HP AC motor and a contactor switch is used to control the blower system. The side curtains are controlled by two 3-phase AC motors, where two contactor switches with an interlock unit is used to control and rotate the motors in both directions. The misters are managed using a solenoid valve and it is controlled by an AC relay switch. There is a water pump, controlled by another contactor switch.

To optically isolate the controllers with the microcontroller, SSR (Solid State Relay) switches with optocouplers were used. The input signal for the SSR switches is either 0 (Logic LOW) or 3.3V (Logic HIGH).

To ensure the safety of the controller, RCCB (Residual Current Circuit Breaker) is used and the micro circuit breakers are used for every electronic device. The complete panel design of the controller unit can be found in the Supporting Information.

The enclosure box was a metal one with IP65 enclosure protection standards, to protect the internal components from dust as well as humidity and water particles. Since this enclosure is metallic, the WiFi connectivity which is used to communicate with the web server can be interfered with. As a solution, the microcontroller unit was taken out from the main controller box and placed into a small plastic enclosure with the same IP65 standards.

**Wiring and power distribution.** Although the communication between sensors was done wirelessly, the power for the sensor nodes and controller node needed to be supplied by a wired media.

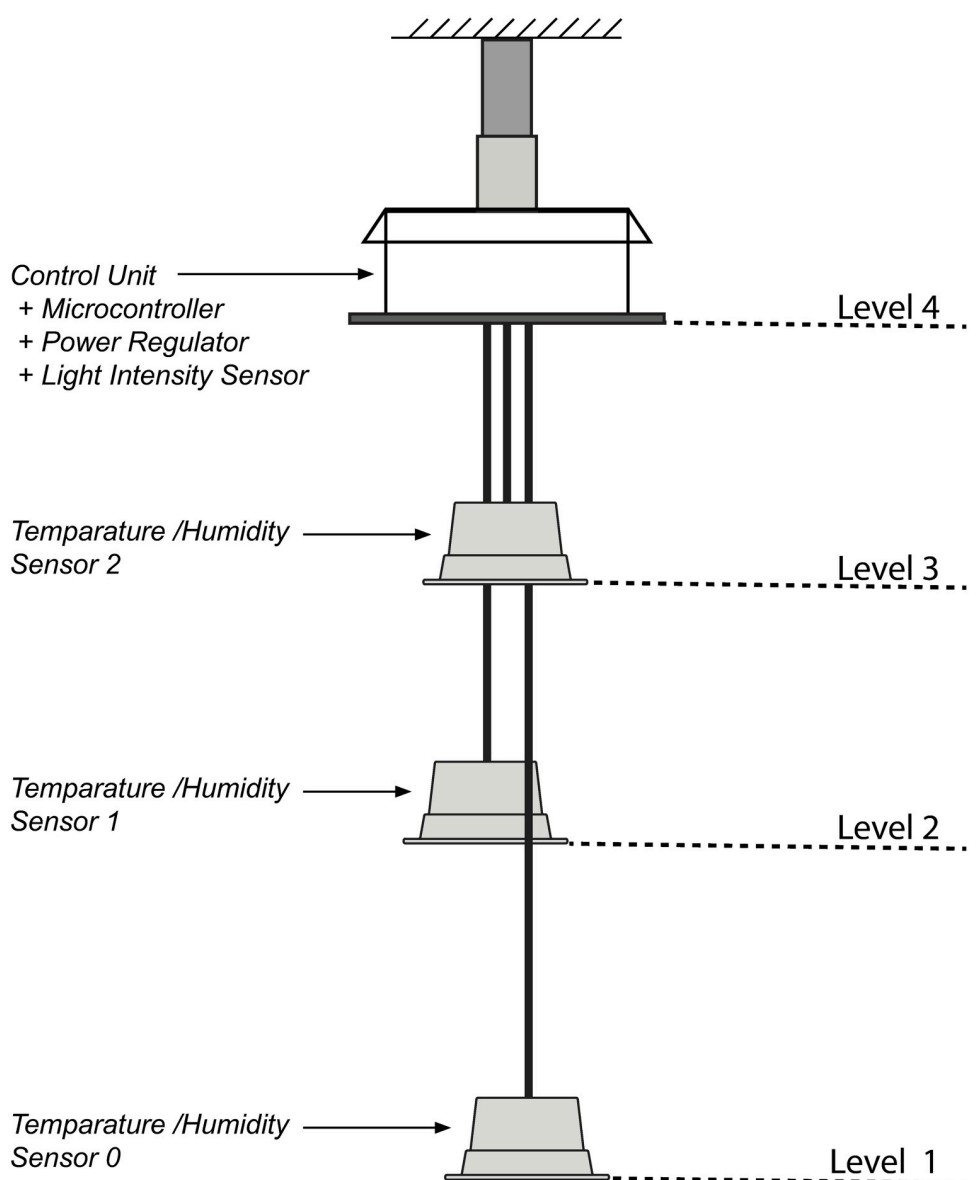

**Fig 1. The arrangement of the sensors in a sensor unit.**

Since there are three-phase devices in the control system, the controller unit was powered by the 3-phase 230V AC line.

Sensor units were designed to work at 3.3V. However, the sensors were located far away from the power distribution board, so it was decided to transmit the power as 12V DC through

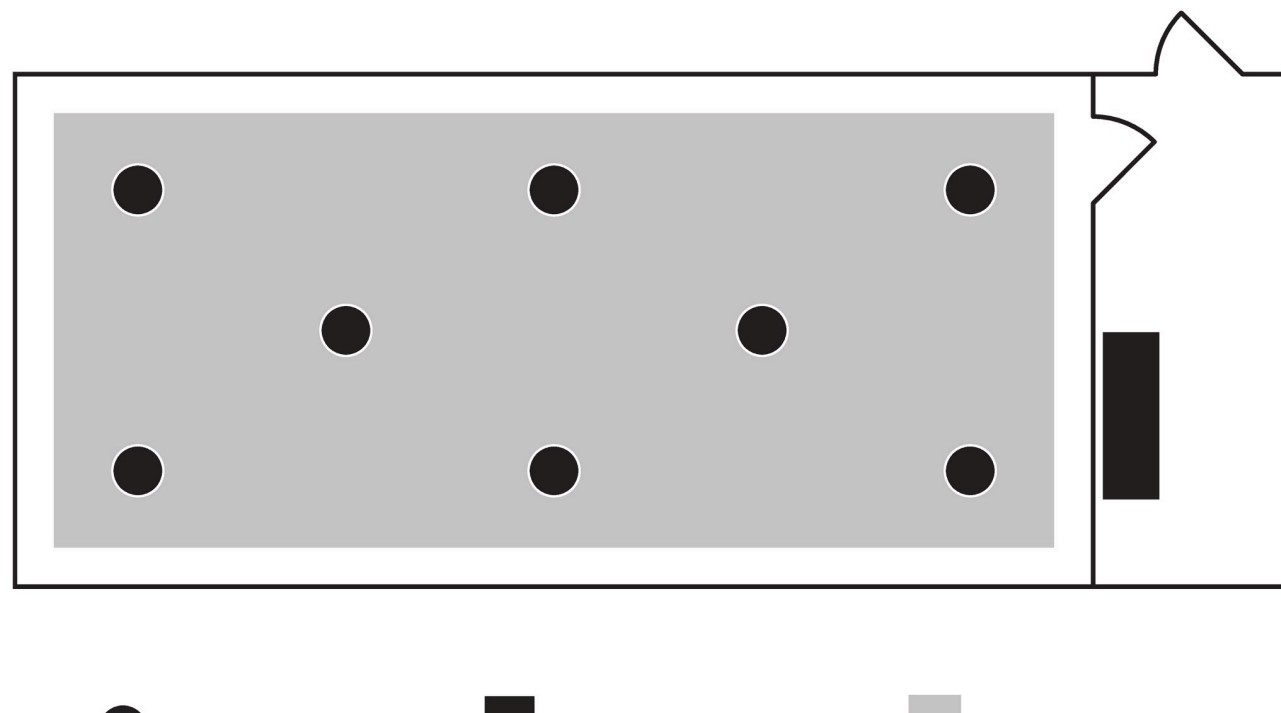

**Fig 2. Suggested arrangement of the sensor nodes inside a rectangular-shaped greenhouse.**

the power line and finally convert the voltage into the 3.3V DC within the sensor unit. With this approach, it is possible to transmit the DC voltage for a long distance, and the voltage drop through the transmission lines will not affect the functionality of the sensor units, as shown in Fig 3.

**Communication.** For communication, Wi-Fi was used, and a Wi-Fi access point was set up in a place nearby the greenhouses, in such a way that every device will get fair Wi-Fi signal coverage. Since both greenhouses were located nearby, and all nodes were within the WiFi router's coverage, no Wi-Fi repeater devices were required. However, if the greenhouses are spread in over a wide geographical area, it is recommended to set up a WiFi mesh network, with a WiFi Gateway and a set of WiFi Repeaters [27].

## System functionality

The system was designed in such a way that the sensor nodes will take sensor readings by a defined frequency and publish them to a specified MQTT topic. A cloud-based web server was developed as an IoT edge server, to look up this published sensor data, pre-process it, and store it in a database for future analytical purposes. Meanwhile, it also checks the incoming sensor readings, executes control logic, and generates control actions such as turning on the blower, turning off the misting system, etc, and publishes those control actions to the MQTT broker (Fig 4).

The controller node is usually subscribed to the controller actions and once the IoT Edge Server publishes a control action, the controller node will receive the action commands. Then it will change the output state of the relevant control according to the input. The actuators in the greenhouse will act according to the control commands received.

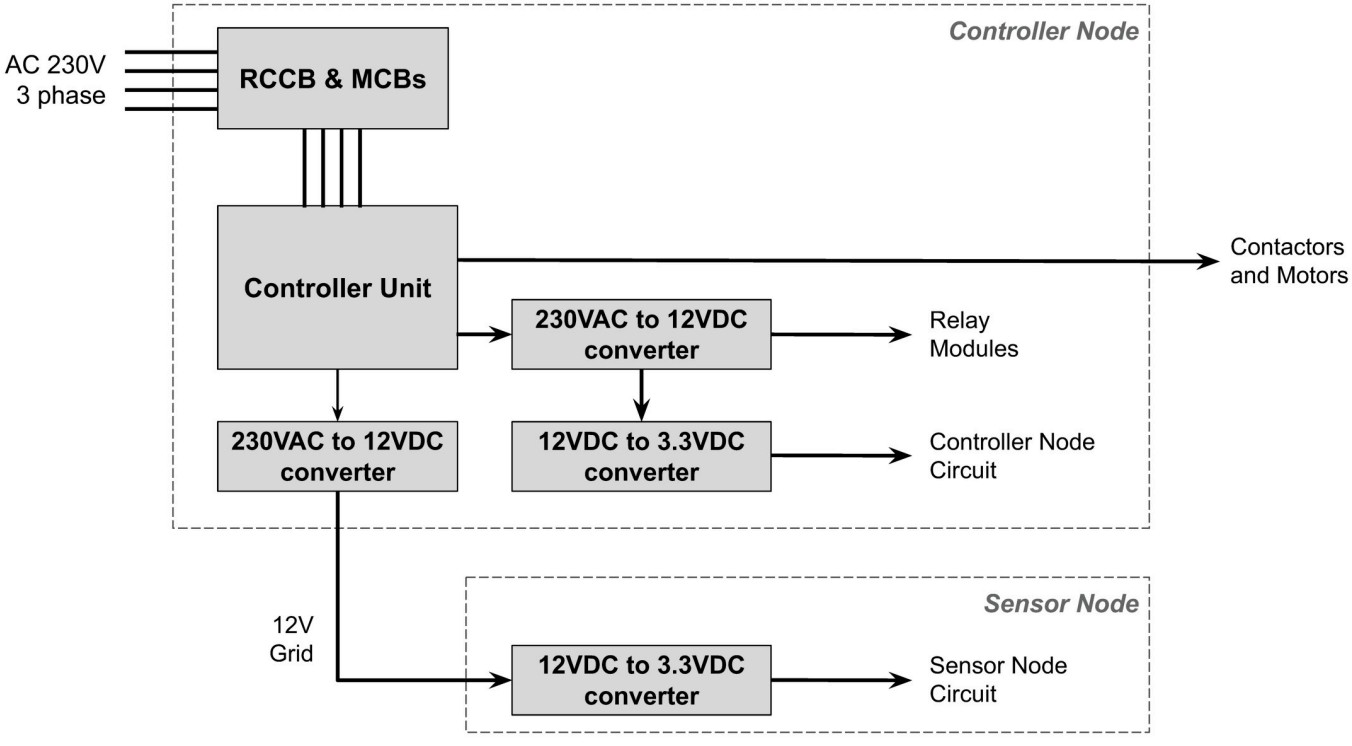

**Fig 3. Power distribution plan of the system.**

Apart from the IoT Edge server, another web server was developed as an IoT dashboard, which can be accessible to the users and see the current state of the system, the previous states as plots, and configure the control logic. This web dashboard also provides a data API, which can be used as a Data Connector to download the sensor readings and use them for statistical computation analysis.

Fig 5 shows several views of the IoT dashboard we implemented with this system. It is capable of handling more than one greenhouse, after defining them with the name and dimensions. Then, it is possible to create and assign the sensor nodes to each greenhouse by defining the location of the sensor node through the web interface. Also, it is possible to select a date and see the variations of the sensor readings using the data visualization tool provided in the web dashboard.

## Cost estimations

A main goal of this research is to introduce a cost-effective controlling and monitoring system for greenhouses. The use of low-cost sensors and controllers makes it more accessible for small-scale farmers to implement greenhouse automation, improving their production capabilities and increasing their profitability. Such systems are also useful for the scientists in resource poor countries to conduct climate change research and adaptation work.

Clear cost comparisons are difficult, since it is subjective to changes based on market prices, regional differences, tax, and import/export policies, etc. Further, the cost of a commercial greenhouse monitoring (and controlling) system varies depending on the features, type of technology, complexity, and target market. Usually, the price ranges from a few thousands to a tens of thousands of dollars. Advanced control systems with IoT features such as real-time remote monitoring, and remote control are significantly more expensive.

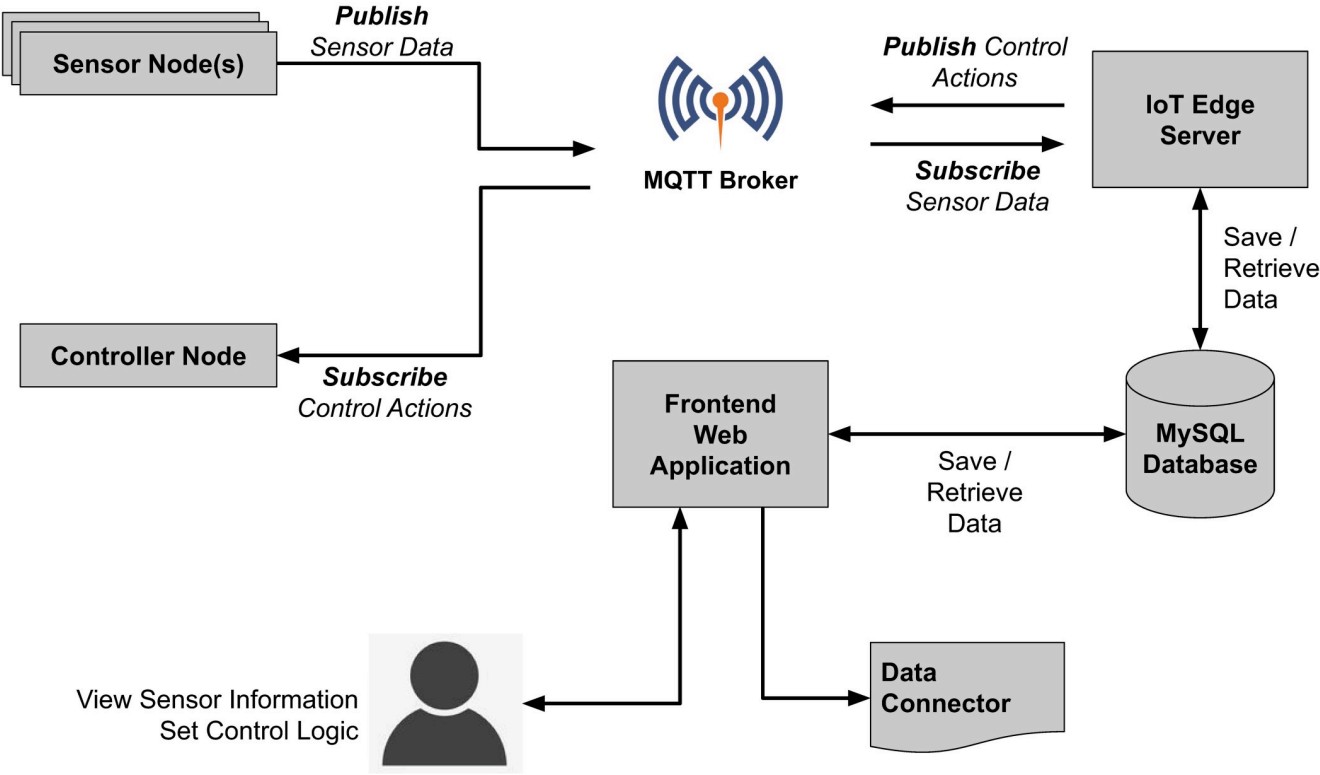

**Fig 4. The overall system architecture of the system.**

As the main approach of this study, we explored the possibility of reducing the cost using the modularity of the design, going for low-cost and low-power sensors with redundant units, and integrating IoT with cloud computing. The followings are the estimated cost per module/unit as of the day the publication is submitted. It is mandatory to have a control unit per polytunnel. About the sensor nodes, one is the minimal requirement per polytunnel, but it is recommended to use at least 3 sensor nodes because taking multiple readings helps to reduce the impact of random variation and gives a more reliable estimate of the true value.

- Controller Unit: $115 − $125

- Sensor Node: $35 − $40

Apart from the hardware cost, there is a cost for the operation, for both hardware and the software. The hardware cost ideally depends on the selection of the controllers, not on the controlling system. For a controller node used during this study, the average power consumption is around 2 to 5 watts, and for a sensor node, it will be less than 1 watt. As for the information, the Blower Fan used during this research contained a 3-Phase 1.11 kW motor, and the misting system had a water pump of single phase 0.37kW, used when the water pressure is not enough. The software and network cost are for the cloud resources used by the system, and for the internet data consumption which is estimated at around $1 to $3 per month, considering the cloud resources and data traffic usage. However, these values may vary with the Internet and Cloud Service Providers selected.

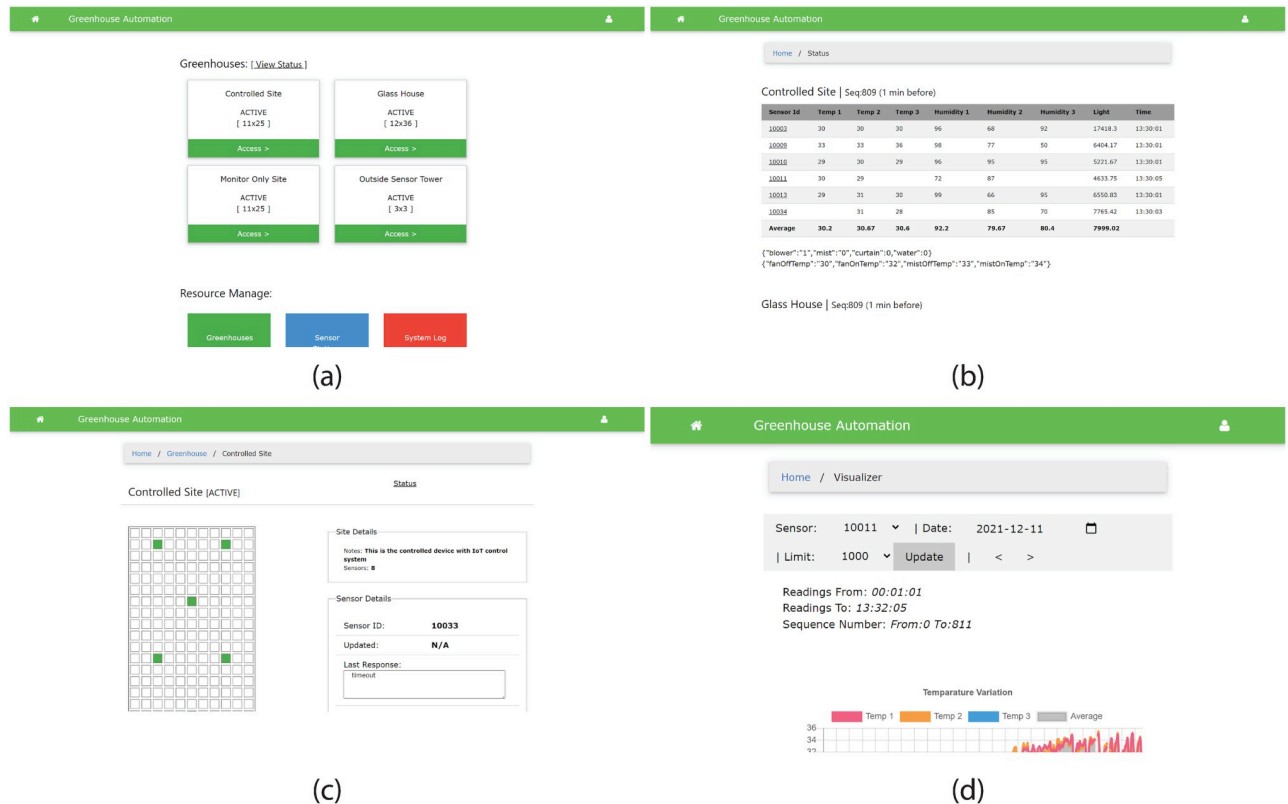

**Fig 5. A few views of the IoT Dashboard: (a) Home screen (b) Current status view (c) Greenhouse sensor arrangement (d) Sensor data visualizer.**

## Test bed setup and experimental data collection

The monitoring system described above was installed in two plastic-covered tunnel greenhouses (poly-tunnels) to compare the system performances, under control and without control. The polytunnels are in Peradeniya, Sri Lanka, and the design and location details can be found in Table 2, and the structural design in Fig 6. The blower fan was oriented to the south of the polytunnel with the controlling system (Fig 7). The tomato hybrid variety, "Sylviana", was planted in polybags in 5 rows.

The temperature, humidity, and light readings of the sensors were recorded at the two poly-tunnels from 16th October 2020 to 5th May 2021 as several trials. It consists of known days with the crop and without the crop to see whether the system performs irrespective of plants. The temperatures 30˚C, and 32˚C were the set points for the blower fan, where the blower fan starts at the temperature > 32˚C and turns off at the temperature < 30˚C. In the same way, the set points used for the misting system were 33˚C and 34˚C. The temperature value was the average of all 24 temperature sensors (8 sensors x 3 levels) in each polytunnel.

## Results

For the experimental data analysis, we used the sensor data collected from the following 2 trials.

- Trial A: 16/10/2020 to 12/01/2021 (89 days) with tomato variety, "Sylviana"

- Trial B: 01/04/2021 to 01/05/2021 (31 days) without plants

**Table 2. Comparison of the polytunnels used for the experiment.**

| | Polytunnel with Control System | Polytunnel without Control System |
|---|---|---|
| Frame | Arch frame | Arch frame with a top vent of 1.5 ft gap |
| Net | • UV-protected clear polythene film (gauge: 300μ) with a double cladding, keeping a 30 cm gap between two claddings. | • UV-protected clear polythene film (gauge size: 300μ) |
| | • Side ventilation was facilitated through an insect-proof net with a mesh size of 40. | • Side ventilation was facilitated through an insect-proof net with a mesh size of 40. |
| Monitoring system | • Temperature and Humidity: 8 locations x 3 levels | • Temperature and Humidity: 8 locations x 3 levels |
| | • Light intensity: 8 sensors | • Light intensity: 8 sensors |
| Controlling system | • Blower motor-based air exhausting system | N/A |
| | • Misting based evaporate cooling system | |
| Location | Peradeniya, Sri Lanka | Peradeniya, Sri Lanka |
| | (7˚ 15' 5.5656" N, 80˚ 35' 38.0364" E) | (7˚ 15' 5.5656" N, 80˚ 35' 38.0364" E) |
| Dimensions (W x L x H) | 50 ft x 20 ft x 12ft | 50 ft x 20 ft x 12ft |

The plots of the meteorological parameter variation were created from the data collected from all the sensors in the two polytunnel systems. The sensor data were collected every 30 seconds and averaged for the minute in the data processing step. We removed the outliers using a z-score and calculated the simple moving averages (SMA) to smooth out the environmental data variations by filtering out the "noise" that occurred due to the malfunctioning of the sensors and sudden environmental changes.

For three representative dates during Trial A, the plots of temperature, humidity, and light intensity variations of the sensor units located in the different places of the polytunnel with a controlling system are shown in Fig 8. The temperature changes measured by the sensor unit numbers 10010, 10012, 10008, and 10009 indicate high variation compared with the other sensor sites. However, all the sensors in the eight locations had similar patterns of temperature fluctuations during the daytime. The results showed that the nighttime temperature at the

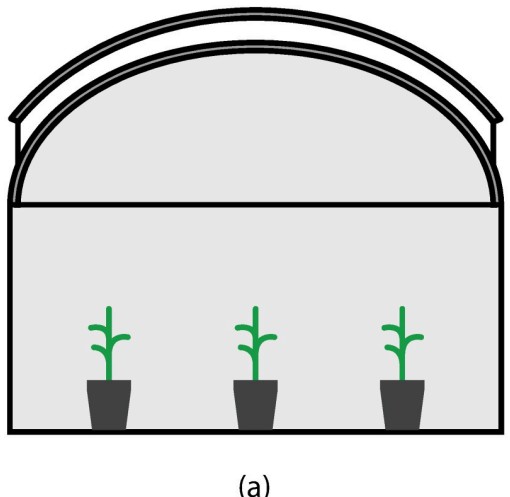
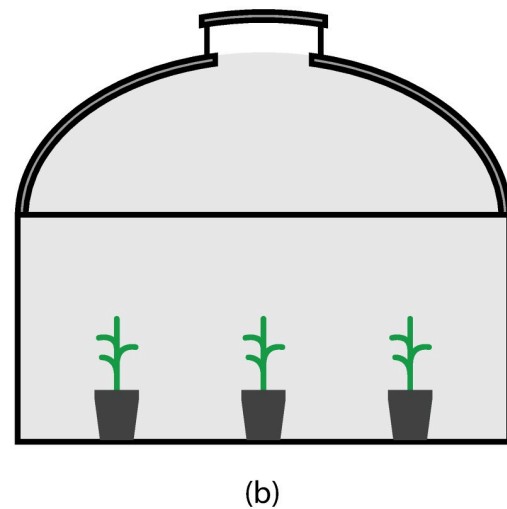

(a) (b)

**Fig 6. Cross Sections of the Polytunnels used during the research (a) Polytunnel with the control system; (b) Polytunnel without the control system.**

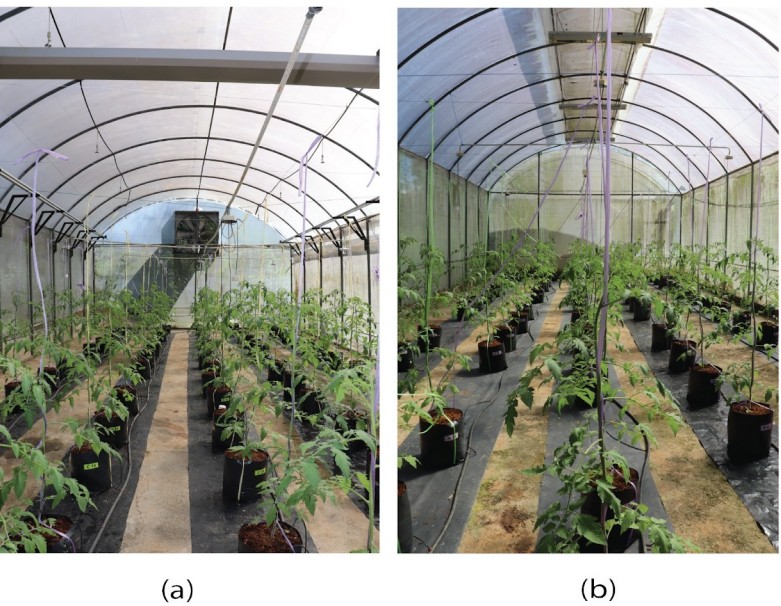

(a)  (b)

**Fig 7. View of the two polytunnels: (a) Polytunnel with the control system; (b) Polytunnel without the control system.**

eight locations was below 25°C. Alongside the temperature inconsistencies, relative humidity distribution in the daytime showed variability among the different sensor locations within the greenhouse. At 15:00–18:00, most of the polytunnel areas experienced a lower favorable temperature while higher temperature variations occurred between 12:00 and 14:00 (Fig 8). The relative humidity distribution in the polytunnel at nighttime showed that a large area of the

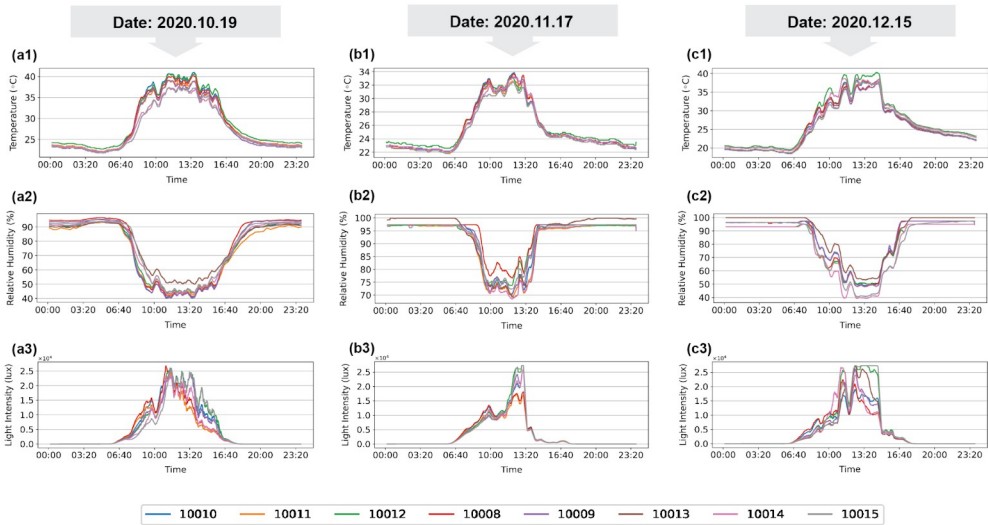

**Fig 8. Variations of meteorological parameters among the eight sensor nodes in the polytunnel with the controlling system for three representative dates. (a1)** Temperature; **(a2)** Relative humidity; **(a3)** Light (RH) on 19 October 2020. **(b1)** Temperature; **(b2)** Relative humidity; **(b3)** Light (RH) on 17 November 2020. **(c1)** Temperature; **(c2)** Relative Humidity; **(c3)** Light (RH) on 15 December 2020.

polytunnel was around 95%. Further, there was a variation in the light intensity levels within the polytunnel due to the surroundings and orientation of the polytunnel.

A representation of temperature and relative humidity variation during the different growth stages is provided in Figs 9 and 10 respectively. During the 50% flowering stage and ripening fruits stage, the temperature in level 2 and level 3 during the daytime indicate similar variation while level 1 fluctuated with lower temperature values compared to the other two levels (Fig 9b and 9d). There is a subtle difference in the temperature variations of the three levels during the vegetative stage and fruit set stage. However, the relative humidity distributions in the three levels show a significant difference.

The polytunnel under control had a lower daytime average temperature variation than the polytunnel without control (Fig 11a). Further, during most days in the trial period, the controlled system maintained the daytime average temperature below/around the set point of 30˚C. The average temperature of the control system in the daytime varied between 30˚C and 25˚C on most days of the trial.

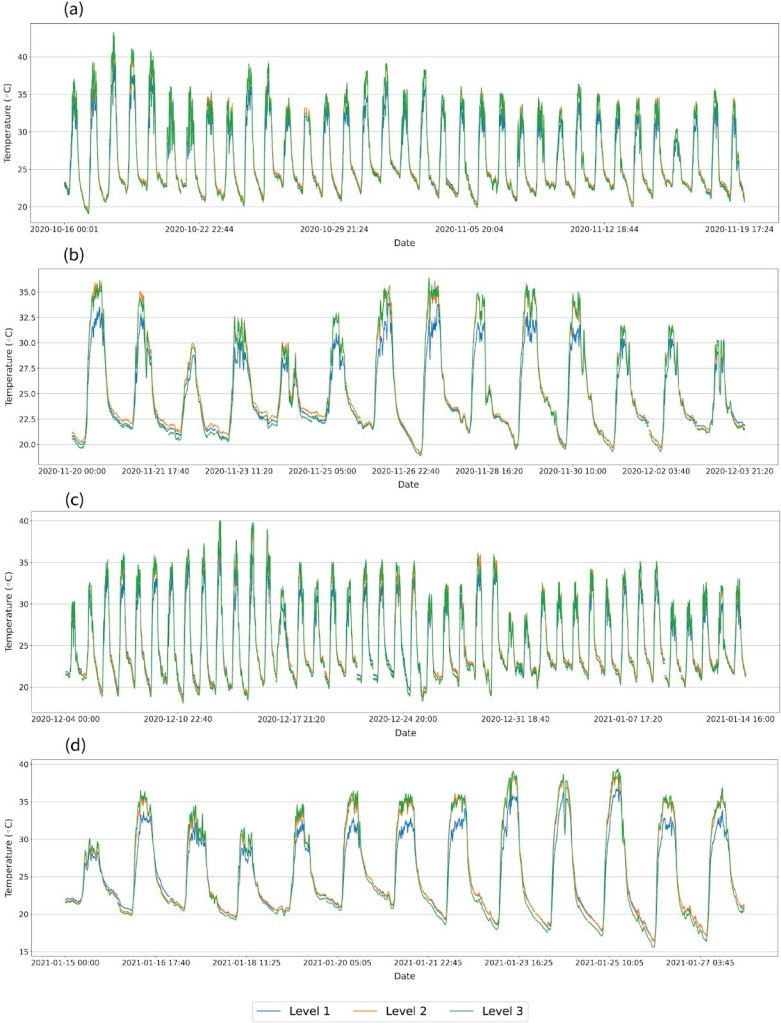

**Fig 9. Temperature variations in the three levels during different tomato growth stages. (a)** Vegetative stage. **(b)** 50% Flowering stage. **(c)** Fruits set the stage. **(d)** Ripen fruit stage. Level 1, Level 2, and Level 3 are 0.4, 1.2, and 2.0 m in height from the floor level respectively.

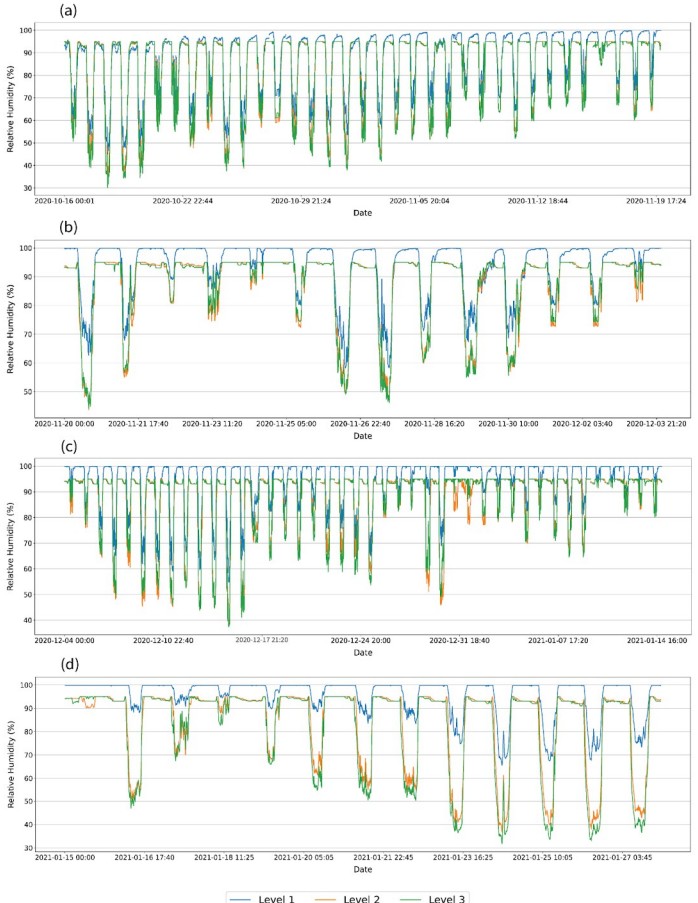

**Fig 10. Relative humidity variations in the three levels during different tomato growth stages. (a)** Vegetative stage. **(b)** 50% Flowering. **(c)** Fruits set the stage. **(d)** Ripen fruit stage. Level 1, Level 2, and Level 3 are 0.4, 1.2, and 2.0 m in height from the floor level respectively.

These results indicate that the implemented system was able to control the temperature within the polytunnel. On sunny days, the blower and misters worked longer time periods to control the temperature within the polytunnel (Table 3). Similarly, the frequencies of the blower and mister activations depended on the outside environmental conditions. Since the controlling system was implemented to maintain the temperature using the misting system, the daytime average humidity level in the polytunnel under control was higher than in the polytunnel without a controlling system (Fig 11b).

Temperature and humidity variation within the polytunnel was compared with and without the tomato plants in Fig 12. The comparison was carried out considering three representative days (sunny, rainy, and cloudy) Since there is a limitation of resources for all four possible experiment setups simultaneously, the experiments with and without plants were done in two different time periods in the available two polytunnels.

Therefore, the side-by-side comparison with exact days with matching weather conditions is not possible, However, we observed that the average daytime, nighttime, and 24-h temperature in the polytunnels with plants were lower than that of the polytunnel without plants (Fig 12a). The temperature difference in the nighttime is lower compared to the daytime. In the controlled polytunnel, it can be observed that the average internal temperatures will be

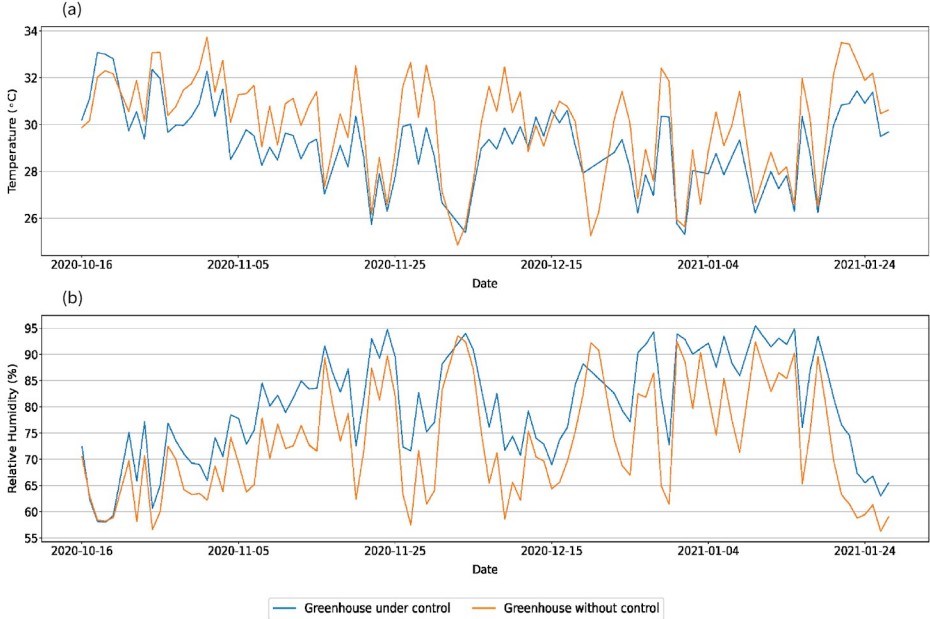

**Fig 11. Daytime average variations of temperature and relative humidity in the two greenhouses throughout the selected trial period. (a)** Temperature **(b)** Relative Humidity.

significantly higher in the without-plant situations compared to the with the plant during the daytime. Moreover, the blower and misters functioned for a longer time to control the temperature in the polytunnel without the plants (Table 3). The blower fan worked from around 09:00 to 15:00 when the plants were established. But when the control system was tested without plants, the blower started around 08:00 and worked until around 17:00 (Table 3).

Further, we performed a statistical comparison among the three systems: the polytunnel with the controlling system, the polytunnel without the controlling system, and the outside environment (Table 4). The ANOVA test [28] was conducted based on the temperature, humidity, and light data of three selected sunny days assuming the systems are independent.

Average temperature values from 06:00 to 10:00 in the three systems show significant differences while the average humidity in the polytunnel without a control system is considerably lower than the other two systems.

From 10.00 to 14.00, the polytunnel systems maintained a lower average temperature compared to the outside. From 14.00 to 18.00, the average temperature of the controlling polytunnel was considerably lower than the outside and the other polytunnel. During the same time, the polytunnel without a controlling system indicated a lower average humidity. The average light intensity values in the two polytunnel systems and the outside were considerably

**Table 3. Maximum time duration (hours) taken to control the temperature in the polytunnel by blower fan and misters.**

| Device | Average time duration (hours) | | | | | |
|---|---|---|---|---|---|---|
| | Sunny day | | Rainy day | | Cloudy day | |
| | With plants | Without plants | With plants | Without plants | With plants | Without plants |
| Blower Fan | 4.15 | 8.65 | 2.03 | 6.80 | 2.27 | 7.22 |
| Misters | 0.43 | 7.40 | 0.08 | 3.18 | 0.13 | 3.62 |

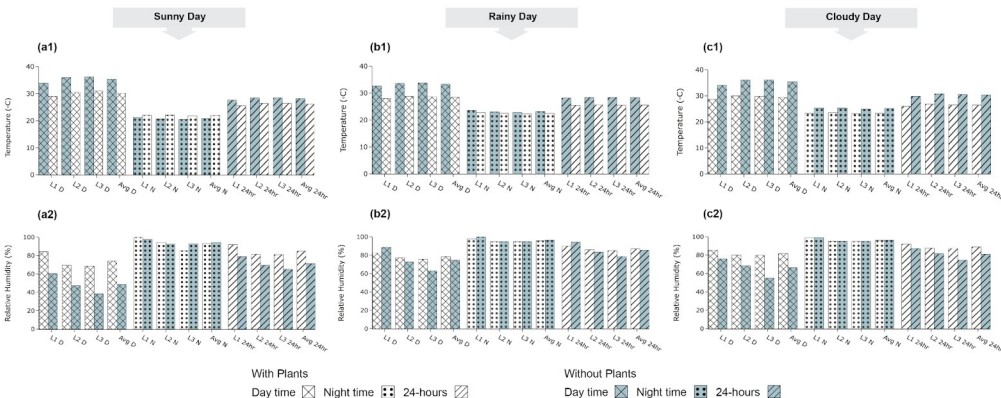

**Fig 12. Average values of temperature and relative humidity with and without plants. (a1)** Temperature; **(a2)** Relative Humidity on a typical sunny day. **(b1)** Temperature; **(b2)** Relative Humidity on a typical rainy day. **(c1)** Temperature; **(c2)** Relative Humidity on a typical cloudy day. D, N, and 24hr are daytime, nighttime, and 24 hours, respectively; Avg denotes average. L1, L2, and L3 are the sensor levels.

different. It may be due to the difference in the polytunnel structures and the variations of the environmental conditions on the selected sunny days.

Another ANOVA test was conducted on the polytunnels considering the systems with plants and without plants (Table 5). According to the results, the average temperature in the polytunnels was higher when the plants were not established. Also, can observe that even for the control system, it is difficult to keep the temperature on the polytunnels below the setpoints without the plant's condition.

All the data used for the above analysis are available from the following data repository as raw data: http://agbc-fe.pdn.ac.lk/datasets/greenhouses/v1/.

## Discussion

In this experiment, we monitored the variation of the temperature and the humidity of the polytunnels with an 8x3 sensor array; where the sensors in 3 different levels, 8 different locations

**Table 4. Environmental conditions of three selective sunny days in different greenhouse conditions.**

| System | Temperature (˚C) | Humidity (%) | Light (lux) |
|---|---|---|---|
| **From 06.00 to 10.00** | | | |
| With controlling system | 27.7 ± 0.6[b] | 80.4 ± 2.8[b] | 4221.5 ± 120.5[b] |
| Without controlling system | 28.9 ± 0.6[a] | 75.5 ± 3.1[c] | 5140.3 ± 242.7[a] |
| Outside | 25.7 ± 0.7[c] | 81.6 ± 1.3[ab] | 4930.4 ± 564.6[a] |
| **From 10.00 to 14.00** | | | |
| With controlling system | 31.2 ± 2.3[b] | 70.8 ± 7.5[ab] | 5196.5 ± 2376.5[c] |
| Without controlling system | 31.2 ± 2.3[b] | 67.6 ± 6.3[ab] | 4014.3 ± 1280.2[c] |
| Outside | 35.2 ± 2.3[a] | 60.± 6 6.1[b] | 15892.3 ± 1793.3[a] |
| **From 14.00 to 18.00** | | | |
| With controlling system | 34.7 ± 1.4[a] | 54.9 ± 4.5[b] | 18787.2 ± 1131.9[a] |
| Without controlling system | 37.0 ± 1.2[a] | 46.8 ± 3.2[c] | 16789.3 ± 678.5[ab] |
| Outside | 35.2 ± 2.4[a] | 60.6 ± 6.1[b] | 15892.3 ± 1793.3[b] |

Values are with mean ± SD of temperature, humidity, and light in different systems. Means denoted by the different letters (a-e) within the same column are significantly different at P < 0.05.

**Table 5. Environmental conditions of five selective sunny days in different greenhouse conditions with/without plants.**

| Condition | Temperature (˚C) | Humidity (%) | Light (lux) |
|---|---|---|---|
| **From 06.00 to 10.00** | | | |
| Polytunnel with the controlling system—with plants | 27.5 ± 0.5[b] | 80.9 ± 4.3[b] | 4351.2 ± 198.6[d] |
| Polytunnel with the controlling system—without plants | 29.4 ± 1.3[a] | 72.4 ± 4.5[cd] | 8237.1 ± 793.0[b] |
| Polytunnel without controlling system—with plants | 28.5 ± 0.9[ab] | 76.2 ± 3.3[c] | 5141.4 ± 602.2[d] |
| Polytunnel without controlling system—without plants | 27.7 ± 1.2[b] | 68.4 ± 2.9[d] | 6817.0 ± 784.6[c] |
| Outside | 25.1 ± 1.2[c] | 85.5 ± 3.9[a] | 9908.2 ± 646.3[a] |
| **From 10.00 to 14.00** | | | |
| Polytunnel with the controlling system—with plants | 35.2 ± 2.4[b] | 54.9 ± 9.7[a] | 18536.7 ± 1194.2[d] |
| Polytunnel with the controlling system—without plants | 41.9 ± 1.4[d] | 41.0 ± 6.8[cd] | 22591.5 ± 678.1[b] |
| Polytunnel without controlling system—with plants | 36.9 ± 0.9[cd] | 47.1 ± 4.0[bc] | 16781.4 ± 1041.9[e] |
| Polytunnel without controlling system—without plants | 39.4 ± 0.6[b] | 38.1 ± 4.5[d] | 21021.2 ± 706.5[c] |
| Outside | 38.1 ± 0.3[bc] | 51.5 ± 7.2[ab] | 26856.6 ± 340.7[a] |
| **From 14.00 to 18.00** | | | |
| Polytunnel with the controlling system—with plants | 30.6 ± 2.6[c] | 71.7 ± 13.3[a] | 4919.5 ± 2333.0[c] |
| Polytunnel with the controlling system—without plants | 37.3 ± 1.0[a] | 52.1 ± 6.0[b] | 7416.3 ± 558.7[b] |
| Polytunnel without controlling system—with plants | 30.7 ± 2.2[c] | 69.3 ± 10.8[a] | 3848.3 ± 1366.4[c] |
| Polytunnel without controlling system—without plants | 34.1 ± 0.8[b] | 51.5 ± 5.2[b] | 8215.8 ± 515.2[b] |
| Outside | 33.6 ± 0.8[b] | 67.3 ± 7.2[a] | 13969.3 ± 612.6[a] |

Values are with mean ± SD of temperature, humidity, and light in different systems. Means denoted by the different letters (a-e) within the same column are significantly different at $P < 0.05$.

which properly cover the entire volume in the polytunnel, and the light intensity received to the plants with 8 sensors for two polytunnels, with the conditions of with and without plants.

Of the two polytunnels used for the experiments, one was with a controlling system, which controls the internal atmosphere conditions by trying to keep the internal temperature below a given setpoint by air exhausting using a blower motor and evaporative cooling using a misting system. In the other polytunnel, no environmental control was done, and just let the natural ventilation to maintain the environmental conditions.

However, one major issue with comparing the environmental variation results between the two polytunnels is the structural difference. Both are arch-framed polytunnels, but the polytunnel without the controlling system has a continuous top vent, while the other polytunnel is without a top vent arrangement. The top vent reduces internal temperature by about 3.6˚C in the same design without a vent under similar environmental conditions. Nevertheless, the control system was able to control the temperature as exacted in the polytunnel without a top vent in which the actual temperature is about 3.6˚C higher than the one with no control system. As such, the actual temperature reduction is higher than the difference that we observed.

For the statistical analysis, the averages were taken from the sensor readings, assuming that the different kinds of weather conditions like sunny days and rainy days will not be caused bias in the readings. For some special comparisons, the days were manually classified into sunny, rainy, and cloudy days using the weather data of the location where the polytunnels were located.

According to the sensor results, it can be seen that the controlling system was able to control the inside temperature under given conditions most of the time. Another important observation was that the plants inside the polytunnels have a bigger contribution to temperature control. During the experiment done without the plants, even the controlling system was unable to keep the internal atmosphere temperature below the setpoints.

The main approach of this research is to introduce low-cost greenhouse monitoring and controlling system. Therefore, the selected sensors were not expensive industrial-grade sensors, so the lifetime and reliability are not guaranteed 100%. However, the redundancy of the sensors (in this experiment, there were 24 sensors per polytunnel) helped to get better, distributed, and fair readings of the environmental conditions compared to one or two expensive sensors, which are taking the measurement of a point or a small region.

The control system used industrial-grade components to interface with the actuators, but the control logic was handled in the specially designed microcontroller unit and an enclosure, which are significantly low cost compared to a PLC-based system. However, the additional electronic circuits and isolation techniques helped to get a similar performance to a PLC control system.

As such, the developed system is also suitable to simulate climate changes in tropical conditions and monitor changes in real-time. The system is suitable to assess the performance of crops in response to set temperatures in terms of gene expression, physiology, anatomy, biochemistry, and quality and quantity of yield. The set parameters can be changed depending on the crop and the objectives of the trails conducted. For example, in the trial reported here and trials repeated with the same tomato variety under the same conditions, the crop growth, and yield were not quite different in the two tunnels (data not shown). However, in another experiment, an ornamental foliage plant behaved differently in two tunnels under the same environmental conditions described here (data not shown). The set points are to be identified for each crop to achieve optimum growth and yield parameters. Further improvements are possible with the integration of a nutrient management system, a light management system, and an image-capturing system.

## Conclusions

Internet of Things (IoT) technology has revolutionized Agriculture, as in many other fields, leading up to smart or precision agricultural systems. To realize the sustainable development goals of agriculture, it is vital that the advantages of smart agriculture become available and affordable to all, with the necessary knowledge and guidance. Therefore, this study was designed to develop low-cost IoT components from basic building blocks, study the performance of the developed systems, and generate real-time experimental data, with and without plants. Low-cost IoT devices developed locally converted existing traditional polytunnels to semi-controlled and monitoring-only polytunnels. Their performances were analyzed and compared with each other based on several matrices while maintaining the planted tomato variety and agronomic practices similar. The developed system performed as expected suggesting the possibility of commercial applications and further research work.

The next stage of this research will be observing the results from an Agronomical perspective and coming up with a better model to identify the best environmental conditions which can maximize the yield of the plants. This will be done using different plant varieties, which require different types of environmental conditions, not only to identify the best varieties to be grown under a controlled environment, but also the optimal conditions to maximize the yield. It will be essential to repeat these experiments in areas with vastly different climates, such as dry, wet, humid, hot, cold, etc., to train the models and have a sufficient amount of data to help the potential users with specific guidance.

From the IoT perspective, this system will be enhanced by the addition of an automated camera system with an image analysis module to map the growth of the plants and identify potential early detection of deceases. Additions of a nutrient management system and a light management system would also allow us to optimize the controlled environment and yield further.

With the generation of adequate amounts of data, it would make it possible to replace the current rule-based control logic with machine learning algorithms to precisely control the environment. To be specific, while a fan and a mister are used to reduce the temperature, they also contribute to increase the relative humidity, respectively. Thus, a properly developed machine learning system will be able to identify how to control the environment more precisely and efficiently. With the integration of data from image analysis, nutrient management, and light management, such a machine learning system could be enhanced even further to identify the most efficient use of resources for a maximum yield.

We expect that such a system would go a long way to achieving sustainable development goals, ensuring food security, and supporting rural development.

## Supporting information

**S1 Fig. PCB design of the sensor node.** ESP32 was used as the microcontroller of the sensor node, since it is a low-cost, low-power SoC (System on Chip), with a dual-core Tensilica Xtensa LX6 microprocessor that works at 160MHz. It is specially designed for wireless IoT sensor applications with necessary modules for GPIO as well as in-built WiFi and Bluetooth functionalities.
(TIF)

**S2 Fig. Controller node design.** High level panel design of the controller, which handles the signal given by ESP32 microcontroller (SoC). There are 8 digital outputs for controlling the devices connected to the controller panel. BLEn, MEn, CAEn, and CBEn are the main output signals and D0-D3 are auxiliary output control signals from the microcontroller. There are 2 optically isolated inputs in the controller unit, to obtain the limit signals from the curtain controller. One input to get the curtain to reach its down-limit signal and one input to get the curtain reached to its up-limit signal. (However, the curtain controller isn't used as a control parameter in this study) Four status indicators were used to indicate the status of the controller node, a power indicator, a WiFi connectivity indicator, a control signal indicator, and an error condition indicator.
(TIF)

## Acknowledgments

Special thanks to all the staff in the Agricultural Biotechnology Center, University of Peradeniya, Sri Lanka, including I.N.S. Dewapriya, P.B.G. Pathirana, R.M.M. Wijerathne, M.G.N.M. Nishshanka, K.M.D.D. Ranaraja, M.M. Kodagoda, E.M.U.L Ekanayake for their immersive support given during crop trials, and maintenance works of the system.

## Author Contributions

**Conceptualization:** K. S. P. Amaratunga, Pradeepa C. G. Bandaranayake, Asitha U. Bandaranayake.

**Data curation:** Sandali Lokuge.

**Funding acquisition:** Pradeepa C. G. Bandaranayake.

**Investigation:** Nuwan Jaliyagoda, Sandali Lokuge.

**Methodology:** Nuwan Jaliyagoda.

**Project administration:** Pradeepa C. G. Bandaranayake.

**Resources:** K. S. P. Amaratunga, Pradeepa C. G. Bandaranayake.

**Software:** Nuwan Jaliyagoda, Sandali Lokuge.

**Supervision:** P. M. P. C. Gunathilake, K. S. P. Amaratunga, W. A. P. Weerakkody, Pradeepa C. G. Bandaranayake, Asitha U. Bandaranayake.

**Validation:** Sandali Lokuge.

**Visualization:** Nuwan Jaliyagoda.

**Writing – original draft:** Nuwan Jaliyagoda, Sandali Lokuge, Asitha U. Bandaranayake.

**Writing – review & editing:** Pradeepa C. G. Bandaranayake.

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
