## [Decision Letter · Decision Letter 0]

25 Jan 2023

PONE-D-22-31496Internet of Things (IoT) for Smart Agriculture: Assembling and assessment of a low-cost IoT systemPLOS ONE

Dear Dr. Bandaranayake,

Thank you for submitting your manuscript to PLOS ONE. After careful consideration, we feel that it has merit but does not fully meet PLOS ONE’s publication criteria as it currently stands. Therefore, we invite you to submit a revised version of the manuscript that addresses the points raised during the review process.

 The suggested references by the reviewers are not required to be included, only add them if deemed necessary.

We look forward to receiving your revised manuscript.

Kind regards,

Sathishkumar V E

Academic Editor

PLOS ONE

Journal Requirements:

4. Our internal editors have looked over your manuscript and determined that it is within the scope of our New Supply Chain Technologies Call for Papers. This collection of papers is curated by Guest Editor Dragan Pamucar (University of Belgrade). The Collection will encompass a diverse and interdisciplinary set of research articles on development and use of Internet of Things devices, in their networking and communication, and in their new knowledge generation capabilities, towards solving challenges in, and related to, supply chain management. Additional information can be found on our announcement page: https://collections.plos.org/call-for-papers/new-supply-chain-technologies/  If you would like your manuscript to be considered for this collection, please let us know in your cover letter and we will ensure that your paper is treated as if you were responding to this call. If you would prefer to remove your manuscript from collection consideration, please specify this in the cover letter.

Reviewers' comments:

Reviewer's Responses to Questions

**Comments to the Author**

1. Is the manuscript technically sound, and do the data support the conclusions?

Reviewer #1: Partly

Reviewer #2: Yes

Reviewer #3: Partly

2. Has the statistical analysis been performed appropriately and rigorously? 

Reviewer #1: Yes

Reviewer #2: Yes

Reviewer #3: N/A

3. Have the authors made all data underlying the findings in their manuscript fully available?

Reviewer #1: Yes

Reviewer #2: Yes

Reviewer #3: Yes

4. Is the manuscript presented in an intelligible fashion and written in standard English?

Reviewer #1: Yes

Reviewer #2: Yes

Reviewer #3: Yes

5. Review Comments to the Author

Reviewer #1: The manuscript has the scholarly importance and it seems look very interesting and to appreciate the authors for their scientific efforts, but need to improve the article is presented in an intelligible fashion and thoroughly check most of word combined (e.g.: polytunnels, Testbed etc)and not fit as per POLS guidelines. These are the following suggestion and comments for author to incorporate before final publication.

1. There is no specific logic mentioned for selecting the area standard of poly-tunnel, in order to place sensor and control the metrological parameters as mentioned under heading Sensor Node design.

2. Under heading communication it is not represents clear meaning “Since both greenhouses were located nearby, no Wi-Fi repeaters were required”. Need to clearify.

3. Conclusion should be

4. Please revise your conclusion part separately into more details. Basically, you should enhance your contributions, limitations, underscore the scientific value added of your paper, and/or the applicability of your findings/results and future study in this session.

Reviewer #2: 1. Numbering specific objectives in the abstract should be removed

2. The research papers demonstrates originality to develop and test IoT system for achieving sustainable goals in agriculture practice.

3. The work is logically designed to develop IoT using basic blocks. However, the word low-cost is not clearly justified... If there is already, an existing IoT system used to monitor greenhouse plants using control parameters like temperature, humidity, etc. I wish the authors could add a section on the IoT cost for smart agriculture and how this system is different.

4. Certain IoT functions like monitoring, controlling are addressed.

5. The polytunnel experiment results showed a reduction of 3.6 oC temperature. What is the change in temperature and humidity for a 24-hr period in the polytunnel without use of sensors? In the control polytunnel, you are using blowers and misters for controlling the temperature. How does this step go to prove the statement of developing a low-cost IoT. These are power-driven devices, which add to the cost of managing the polytunnel.

6. The effect of changes in environment conditions of polytunnel and plant yield should be highlighted.

Reviewer #3: The paper entitled “Internet of Things (IoT) for Smart Agriculture: Assembling and assessment of a low-cost IoT system”, although is utterly interesting and addresses to the readership of the Journal, appears to have some issues in which the authors should put more effort:

1. The authors should reconsider the title of their manuscript taking into account the fact that it mostly refers to controlled agricultural environments such as greenhouses and not open arable areas.

2. The subject of research is not entirely new, since several works have been published on it. Although the authors cite a relatively high number of references they fail to substantiate the novelty as well as the scientific contribution of their work. To this end, the authors should rework the content of the introduction section by enriching it with a number of references targeting on the subject of: (a) the current trends and challenges in the Deployment of IoT Technologies for climate smart facility agriculture as for instance the work in:

https://doi.org/10.1504/IJSAMI.2019.101673

and (b) the IoT technologies in smart greenhouses such as for instance the work in:

https://doi.org/10.1016/j.compag.2022.106993.

Finally, additional justifications should be included in the final paragraph of the introduction

3. The methodology which is followed in this work as well as the deriving results are not clearly defined. To be more precise, the system design as well as the results seem to be rather superficial whereas the lack of Figures in the entire manuscript does not assist the work of this reviewer in comprehending critical points of the methodology as the system architecture for instance. As for the results, the authors include a number of tables with data regarding temperature, humidity, etc., they do not however present tables with data regarding the performance of the system such as response time, number of requests, energy consumption, etc. This issue has to be attended.

4. The conclusion section is rather short and generic. The findings of the research and their implications should be discussed in the broadest context possible while limitations on the subject of research should be also further highlighted. A discussion section would aid in this direction.

5. The paper is written in good English however some spell and grammar checking is required.

To sum up, although the subject of this work is quite interesting, the manuscript needs to be majorly revised in order to be published.

6. PLOS authors have the option to publish the peer review history of their article (what does this mean?). If published, this will include your full peer review and any attached files.

Reviewer #1: **Yes: **Dr. Dharmendra

Reviewer #2: **Yes: **Gurumurthy B Ramaiah

Reviewer #3: **Yes: **Prof. Konstantinos G. Arvanitis

---

## [Author Response · Author response to Decision Letter 0]

28 Mar 2023

[This is already submitted as a PDF attachment]

General Comments

In your Methods section, please provide additional information regarding the permits you obtained for the work. Please ensure you have included the full name of the authority that approved the field site access and, if no permits were required, a brief statement explaining why.

According to the current rules and regulations in the country, no specific permission or approvals were required from any organization to conduct this research. (Mentioned in Page 5)

Please note that PLOS ONE has specific guidelines on code sharing for submissions in which author-generated code underpins the findings in the manuscript. In these cases, all author-generated code must be made available without restrictions upon publication of the work. Please review our guidelines at https://journals.plos.org/plosone/s/materials-and-software-sharing#loc-sharing-code and ensure that your code is shared in a way that follows best practice and facilitates reproducibility and reuse.

We are also happy to make our works open-source. A part of the technical information comes under the support information document in the submission. The rest of the implementation details with source-code is openly available in the following code repository, https://github.com/cepdnaclk/Smart-Agriculture-a-low-cost-IoT-system-for-polytunnels

In your Data Availability statement, you have not specified where the minimal data set underlying the results described in your manuscript can be found. PLOS defines a study's minimal data set as the underlying data used to reach the conclusions drawn in the manuscript and any additional data required to replicate the reported study findings in their entirety. 

We published our datasets to the public and the access details to the datasets mentioned on the Page 21

Reviewer #1 

The manuscript has the scholarly importance and it seems look very interesting and to appreciate the authors for their scientific efforts, but need to improve the article is presented in an intelligible fashion and thoroughly check most of word combined (e.g.: polytunnels, Testbed etc) and not fit as per POLS guidelines. These are the following suggestion and comments for author to incorporate before final publication.

We go through your review comment, and update the term ‘Testbed’ with ‘Test bed’ (Page 14). However, ‘polytunnel’ is considered a single word in most of the places in the literature, so we kept it as it is.

1. There is no specific logic mentioned for selecting the area standard of poly-tunnel, in order to place sensor and control the metrological parameters as mentioned under heading Sensor Node design.

As per your comment, we further explained the design designs on both the sensor node and controllers node under the section “System design” (Page 7-9)

2. Under heading communication it is not represents clear meaning “Since both greenhouses were located nearby, no Wi-Fi repeaters were required”. Need to clearify.

We have addressed that part and included more generalized reasoning for the above statement in the Page 11

3. Conclusion should be

4. Please revise your conclusion part separately into more details. Basically, you should enhance your contributions, limitations, underscore the scientific value added of your paper, and/or the applicability of your findings/results and future study in this session.

The conclusion section was revised as suggested (Page 24-26).

Reviewer #2

1. Numbering specific objectives in the abstract should be removed

Revised as suggested (Page 1).

2. The research papers demonstrates originality to develop and test IoT system for achieving sustainable goals in agriculture practice.

3. The work is logically designed to develop IoT using basic blocks. However, the word low-cost is not clearly justified... If there is already, an existing IoT system used to monitor greenhouse plants using control parameters like temperature, humidity, etc. I wish the authors could add a section on the IoT cost for smart agriculture and how this system is different.

Since it is a little difficult to directly include the prices with references since it is subjective to change from time to time as well as not possible according to the journal policies. 

However, a new section was added under, “Materials and Methods”, as “Cost Estimations”, where we discuss the cost-related content and justifications, especially the setup cost and the operational cost (Page 12-14). 

4. Certain IoT functions like monitoring, controlling are addressed.

5. The polytunnel experiment results showed a reduction of 3.6 oC temperature. What is the change in temperature and humidity for a 24-hr period in the polytunnel without use of sensors? In the control polytunnel, you are using blowers and misters for controlling the temperature. How does this step go to prove the statement of developing a low-cost IoT. These are power-driven devices, which add to the cost of managing the polytunnel.

There is no polytunnel without sensors. Both polytunnels consist of sensors while one of them also has a control system.

In this study, our primary target was to reduce the cost of setup. The operational cost mainly depends on the control devices, in this case, the blowers and misters. We didn’t compare the operational cost of the system, but it is a good suggestion to take consider in the next phase of this research.

6. The effect of changes in environment conditions of polytunnel and plant yield should be highlighted.

Several crop cycles are completed so far and there is a comprehensive dataset was collected on crop growth, physiology, gene expression, agronomy, etc. The newly developed system presented in this manuscript was utilized for the experiments. The objective was to show the adaptation of the selected variety (tomato) to climate change. Therefore, the data set and analysis do not match the theme of this manuscript. That is the reason to only present the system and its performance in this manuscript.

Reviewer #3

The paper entitled “Internet of Things (IoT) for Smart Agriculture: Assembling and assessment of a low-cost IoT system”, although is utterly interesting and addresses to the readership of the Journal, appears to have some issues in which the authors should put more effort:

1. The authors should reconsider the title of their manuscript taking into account the fact that it mostly refers to controlled agricultural environments such as greenhouses and not open arable areas.

We also agree with the suggestion, and therefore we have updated the title of our study to, “Internet of Things (IoT) for Smart Agriculture: Assembling and assessment of a low-cost IoT system for polytunnels”.

2. The subject of research is not entirely new, since several works have been published on it. Although the authors cite a relatively high number of references they fail to substantiate the novelty as well as the scientific contribution of their work. To this end, the authors should rework the content of the introduction section by enriching it with a number of references targeting on the subject of: 

(a) the current trends and challenges in the Deployment of IoT Technologies for climate smart facility agriculture as for instance the work in: https://doi.org/10.1504/IJSAMI.2019.101673

(b) the IoT technologies in smart greenhouses such as for instance the work in: https://doi.org/10.1016/j.compag.2022.106993.

Finally, additional justifications should be included in the final paragraph of the introduction

We have updated the content to reflect the objective we had, to work on designing a low-cost and low-power, IoT-based network of devices, which can monitor the environment inside a polytunnel with 3-dimensional coverage on temperature and humidity variations, as well as 2-dimensional coverage of the light intensity received. The idea on 3-dimensional and 2-dimensional measurement coverage of polytunnels is not found yet in the literature as per our knowledge. 

Further, we compared the result we obtained with several different environment setups and represented in graphically, which will be helpful for researches who are willing to go further in the research domain.

3. The methodology which is followed in this work as well as the deriving results are not clearly defined. To be more precise, the system design as well as the results seem to be rather superficial whereas the lack of Figures in the entire manuscript does not assist the work of this reviewer in comprehending critical points of the methodology as the system architecture for instance. As for the results, the authors include a number of tables with data regarding temperature, humidity, etc., they do not however present tables with data regarding the performance of the system such as response time, number of requests, energy consumption, etc. This issue has to be attended.

As mentioned in the introduction section, the main research gap in this domain is not having sufficient real data to come up with innovative solutions. As a solution to that, we decided to share the experience we obtained in designing a system that can collect the required data, with all the design decisions that may be useful to continue. The results section was dedicated to showing the findings of our experiments with the observations we had. 

When come to performance, the system we developed is not a hard real-time system, so the response time and the performance of the system based on a number of requests haven’t a significant role. Those topics were already studied under the domain of IoT, and we followed the industry standards developed according to the outcomes of those studies while designing the system. 

4. The conclusion section is rather short and generic. The findings of the research and their implications should be discussed in the broadest context possible while limitations on the subject of research should be also further highlighted. A discussion section would aid in this direction.

We improved the Discussion and Conclusion sections as suggested (Page 22-26)

5. The paper is written in good English however some spell and grammar checking is required.

We have revised the manuscript as suggested.

* Please note that the page numbers mentioned here are from the revised manuscript, without tracked changes.

---

## [Decision Letter · Decision Letter 1]

9 May 2023

Internet of Things (IoT) for Smart Agriculture: Assembling and assessment of a low-cost IoT system for polytunnels

PONE-D-22-31496R1

Dear Dr. Bandaranayake,

We’re pleased to inform you that your manuscript has been judged scientifically suitable for publication and will be formally accepted for publication once it meets all outstanding technical requirements.

Kind regards,

Sathishkumar V E

Academic Editor

PLOS ONE

Additional Editor Comments (optional):

Reviewers' comments:

Reviewer's Responses to Questions

**Comments to the Author**

1. If the authors have adequately addressed your comments raised in a previous round of review and you feel that this manuscript is now acceptable for publication, you may indicate that here to bypass the “Comments to the Author” section, enter your conflict of interest statement in the “Confidential to Editor” section, and submit your "Accept" recommendation.

Reviewer #1: All comments have been addressed

Reviewer #2: All comments have been addressed

Reviewer #3: All comments have been addressed

2. Is the manuscript technically sound, and do the data support the conclusions?

Reviewer #1: Yes

Reviewer #2: Yes

Reviewer #3: Yes

3. Has the statistical analysis been performed appropriately and rigorously? 

Reviewer #1: Yes

Reviewer #2: Yes

Reviewer #3: N/A

4. Have the authors made all data underlying the findings in their manuscript fully available?

Reviewer #1: Yes

Reviewer #2: Yes

Reviewer #3: Yes

5. Is the manuscript presented in an intelligible fashion and written in standard English?

Reviewer #1: Yes

Reviewer #2: Yes

Reviewer #3: Yes

6. Review Comments to the Author

Reviewer #1: Author has respond and explained all those points which I have raised in my previous review therefore as per my point of view it looks good.

Reviewer #2: 1. The study presents unique research experimental data

2. The analyses are performed with good and high technical standard

3. Conclusions are presented in an appropriate fashion and are supported by the data.

4. The article is presented in an intelligible fashion and is written in standard English.

5. The research meets all applicable standards for the ethics of experimentation and research integrity.

6. The article adheres to appropriate reporting guidelines and community standards for data availability.

Reviewer #3: No further comments. The revision was satisfactory. The authors have taken into account my queries and comments.

7. PLOS authors have the option to publish the peer review history of their article (what does this mean?). If published, this will include your full peer review and any attached files.

Reviewer #1: **Yes: **Dr. Dharmendra

Reviewer #2: **Yes: **Gurumurthy B Ramaiah

Reviewer #3: **Yes: **Prof. Konstantinos G. Arvanitis

---

## [Editor Report · Acceptance letter]

17 May 2023

PONE-D-22-31496R1 

Internet of Things (IoT) for Smart Agriculture: Assembling and assessment of a low-cost IoT system for polytunnels 

Dear Dr. Bandaranayake:

I'm pleased to inform you that your manuscript has been deemed suitable for publication in PLOS ONE. Congratulations! Your manuscript is now with our production department. 

Kind regards, 

on behalf of

Dr. Sathishkumar V E 

Academic Editor

PLOS ONE